Piecing together the biogeographic history of Chenopodium vulvaria L. using botanical literature and collections

Groom Quentin J. quentin.groom@br.fgov.be
Botanic Garden Meise , Bouchout Domain, Meise , Belgium
Esler Karen
Electronic publication date: 2015 Jan 8
Publication date: 2015
Volume: 3
Electronic Location ID: e723
Received 2014 Oct 13; Accepted 2014 Dec 22
Copyright: © 2015 Groom
Copyright year: 2015
Copyright holder: Groom
License: This is an open access article distributed under the terms of the Creative Commons Attribution License, which permits unrestricted use, distribution, reproduction and adaptation in any medium and for any purpose provided that it is properly attributed. For attribution, the original author(s), title, publication source (PeerJ) and either DOI or URL of the article must be cited.
License URL: https://creativecommons.org/licenses/by/4.0/

Keywords: Text analysis, Habitat change, Introduction pathways, Bioclimatic modelling, Distribution, Naturalisation, Herbarium specimens

Funding: European Commission under the 7th Framework Programme 312848 This work has been supported by the pro-iBiosphere project funded by the European Commission under the 7th Framework Programme, grant agreement number 312848. The funders had no role in study design, data collection and analysis, decision to publish, or preparation of the manuscript.

==============================
This study demonstrates the value of legacy literature and historic collections as a source of data on environmental history. Chenopodium vulvaria L. has declined in northern Europe and is of conservation concern in several countries, whereas in other countries outside Europe it has naturalised and is considered an alien weed. In its European range it is considered native in the south, but the northern boundary of its native range is unknown. It is hypothesised that much of its former distribution in northern Europe was the result of repeated introductions from southern Europe and that its decline in northern Europe is the result of habitat change and a reduction in the number of propagules imported to the north. A historical analysis of its ecology and distribution was conducted by mining legacy literature and historical botanical collections. Text analysis of habitat descriptions written on specimens and published in botanical literature covering a period of more than 200 years indicate that the habitat and introduction pathways of C. vulvaria have changed with time. Using the non-European naturalised range in a climate niche model, it is possible to project the range in Europe. By comparing this predicted model with a similar model created from all observations, it is clear that there is a large discrepancy between the realized and predicted distributions. This is discussed together with the social, technological and economic changes that have occurred in northern Europe, with respect to their influence on C. vulvaria.

Introduction

Legacy biodiversity literature is a potential source of useful information on the past distributions of organisms. While these texts have always been available in academic libraries, their accessibility and discoverability has been significantly enhanced by projects such as the Biodiversity Heritage Library (www.biodiversitylibrary.org) and other online digital sources. The ability to search a whole corpus of historical literature for a Latin name of an organism dramatically increases the accessibility of this information and makes literature searches possible that once would have been unfeasible. In parallel, the widespread digital imaging of herbarium specimens and transcription of their labels has also made these data considerably more accessible, which, combined with historic literature, has created a large pool of information from which the phytogeographic historian can draw evidence (Vellend et al., 2013).

Chenopodium vulvaria L. is a small, inconspicuous species that grows largely in places disturbed by humankind. It is not remarkable morphologically, but it is nonetheless distinctive due to its foul smell, which is described as similar to that of rotten fish. Its distinctiveness makes it particularly suited to a study using historic literature, because there is less concern that published accounts refer to other species as a result of misidentification.

C. vulvaria is currently a red-listed species in several countries including Sweden (www.artfakta.se), the United Kingdom (Cheffings et al., 2005), Belgium (Kestemont, 2010), Luxembourg (Colling, 2005), Czech Republic (Grulich, 2012) and some regions of France (Ferrez, 2005). In Great Britain, Ireland and Flanders, comparisons of atlas data show that C. vulvaria is in severe decline (Preston, Pearman & Dines, 2002; Van Landuyt et al., 2006). In contrast, it has naturalised in California (Calflora, 2014), Argentina (Planchuelo, 1975; Giusti, 1997), Chile (Boelcke et al., 1985) and Australia (Atlas of Living Australia, 2014).

C. vulvaria is widespread in countries bordering the Mediterranean and eastward to Afghanistan and Mongolia (Jalas & Suominen, 1980; Meusel, Jäger & Weinert, 1992). Yet it is clear from historical literature and specimens that it was common in parts of northern Europe during the 18th and 19th centuries. Turner (1548) wrote “It groweth muche aboute the walles in Bon in Germany”; Bucher (1806) wrote in the Flora of Dresden “An den strassen der vorstadt und sonst gemein” translated as “By the streets of suburbs and usually common”; Curtis (1777) stated “This species is very common in the neighbourhood of London…” and Hooker (1821), in his flora of Scotland, describes it as “frequent”. Lejeune & Courtois (1828) described its distribution in Belgium as “…per totum Belgium passim”, which can be translated as “everywhere throughout Belgium” and in France it was described as “commun” and “trés commune” (i.e., “common” and “very common”) on specimens from Narbonne (1846, P04922786) and Paris (1819, P05292341) in the Museum National d’Histoire Naturelle.

The native distribution of C. vulvaria is unknown and its long association with anthropogenic disturbance makes this impossible to determine. Floras in Northern and Central Europe variously describe it either as a native or an archaeophyte, though the evidence for categorizing it in either category is slim and is probably based on the anthropogenic habitats that C. vulvaria often inhabits.

Many other members of the Amaranthaceae  live in disturbed, nutrient rich habitats and may be halophytic. C. vulvaria itself is often found in disturbed, eutrophic and coastal habitats. In general, species of such habitats are increasing and spreading in northern Europe (Wróbel, Tomaszewicz & Chudecka, 2006; Van Landuyt et al., 2008; Smart et al., 2003; Šerá, 2011; Groom, 2013). So, at face value, C. vulvaria appears well adapted to modern habitats in Europe and yet it has declined.

One possible explanation for its apparent decline in northern and central Europe may be a misunderstanding of its former occurrence, its presence in the north being the result of propagule pressure from its heartland in southern Europe, constantly reinforcing the introduced populations in the north. One or many introduction pathways may have existed that delivered C. vulvaria seed outside of its normal range and these pathways have since reduced in importance, causing a collapse in the population. Another possible explanation is change to its former habitat, though the details of its ecology are too poorly known to understand what these changes may have been.

For non-woody plants there are few sources of data to examine recent biogeographic change. Palynology and the study of archaeological remains can be useful, but many species do not have sufficiently distinctive characters to identify them from their remains. In these cases, historical literature and collections may be the only sources of data on their former habitats and locations.

Given the shortage of data, an alternative approach, widely used to model the potential distribution of organisms, is bioclimatic modelling. Many studies have used observations from the known native range of a species to extrapolate its potential invasive range (e.g., Macfadyen & Kriticos, 2012). In ecological theory the potential bioclimatic range is generally considered to be larger than the realized distribution as a consequence of additional non-climatic limitations to distribution, such as edaphic factors (Araujo & Peterson, 2012). However, in the case of C. vulvaria the native range is not known, and frequent non-persistent introductions mean that the realized distribution predicted from observations may be larger than its true bioclimatic range. For C. vulvaria the location of naturalisation in Australia, North America and South America might be a clearer indication of its bioclimatic range than within Europe, where it is hard to distinguish established from casual occurrences. Assuming that this species is well established and stable in its alien range, we can use the known naturalised range to model the climate envelope and extrapolate this to Europe to identify the areas where the climate is suitable for C. vulvaria. In this manner, we can indicate those places where this species has been observed but is unlikely to be persistent. A similar approach has been used by Zhu et al. (2012) to predict the source of the invasive insect Halyomorpha halys (Stål, 1855) and has been used by others to model the historical distributions of trees (Benito-Garzón, Sánchez de Dios & Sáinz Ollero, 2008). This approach assumes niche conservatism, but such conservatism is apparently commonplace and perhaps typical of invasive terrestrial plant species (Townsend Peterson, 2011; Petitpierre et al., 2012).

My hypothesis is that C. vulvaria was formally more abundant in northern Europe and its current decline is the result of changes in the introduction pathways and loss of habitat. In this study I draw on botanical literature and specimens to identify habitat change and historic introduction pathways. I use text analysis of habitat descriptions to demonstrate how its habitat has changed over the past 200 years, and I use bioclimatic niche modelling to contrast the realized range within Europe with the projected range based upon naturalised occurrences outside Europe.

Methods

Observation and specimen details were collected in a Common Data Model (CDM) database which is the central component of the EDIT Platform for Cybertaxonomy (Ciardelli et al., 2009; Berendsohn et al., 2011). Two methods were used to extract observations from literature: either XML markup or direct data entry. Digitised treatments were marked up with XML using the GoldenGate editor (http://plazi.org/?q=GoldenGATE, Sautter, Böhm & Agosti, 2007), uploaded to the PLAZI taxonomic treatment repository (plazi.org) and imported to the CDM database. Alternatively, the observation details were copied from the treatment and entered manually into the CDM database using the EDIT Taxonomic Editor (Ciardelli et al., 2009). Observations were collected from the biodiversity literature by reading the Biodiversity Heritage Library corpus systematically after searching for C. vulvaria L. and its synonyms C. foetidum Lam., C. olidum Curt., Atriplex vulvaria Crantz and Vulvaria vulgaris Bubani. Other published observations were gathered from publications in the Library of the Botanic Garden Meise. A list of the sources of observations of C. vulvaria is available in File S1. A complete survey of non-digitised literature is unfeasible, but there was an effort to check multiple floras of every European country and any other country with a temperate climate suitable for C. vulvaria.

Digitised observation data were also gathered from databases, primarily from the Global Biodiversity Information Facility (GBIF) (data.gbif.org, accessed 08 Nov 2013; see File S2), but also from the Atlas of Living Australia (2014), the Botanical Society of Britain and Ireland (2013) and Herbaria United (2013). Scientific articles and websites containing observations were also discovered using search engines (scholar.google.be; google.be). Data were standardised and imported directly into the CDM database.

Specimen data were gathered from herbaria by transcription of label information. Specimens from 31 herbaria are included in the study, and their abbreviations follow the Index Herbariorum (http://sciweb.nybg.org/science2/IndexHerbariorum.asp): ABS, ALA, B, BBB, BC, BIRM, BM, BR, C, CONC, FABR, FRU, HFLA, K, L, LISU, M, MANCH, MW, NYS, P, PRC, RNG, SO, SLBI, SOM, SOMF, UBC, UC, WAG, WU and others contributing data to GBIF (Herbaria names in File S2). Many other herbaria and herbarium catalogues were searched without finding specimens, and several herbaria were contacted and either contained no specimens or did not respond. Undoubtedly there are more specimens and observations of C. vulvaria to be discovered, but I believe these to be a representative sample and a large proportion of those that exist. Undated specimens were not used in the study; however, it is usual for published observations to be undated and therefore the publication date was used instead. In studying biographical information of collectors it is clear that most undated observations in old floras are within 35 years of the publication date, and authors tend to provide dates when those dates are a long time before the publication date. In total, 2456 observations were collected from specimens and literature. These data span 465 years from 1548 to 2013, though there are only two observations from the 16th century, two from the 17th century and nineteen from the 18th century.

Text analysis of habitats

The text describing the habitat of C. vulvaria was collected from 104 floras, 33 scientific articles, 119 specimens and 5 websites, covering the years 1787 to 2014. The texts were written in 12 languages, English (35%) German (20%), French (17%), Latin (12%), Dutch (4%), Italian (3%), Portuguese (3%), Spanish (3%), Hungarian (1%), Danish (1%), Catalan (1%) and Czech (<1%). Each description was broken down into tokens consisting of either single words or short phrases describing a single aspect of the habitat. Thus the description “In Straßen, an Häusern, Stallungen, Düngerstätten” was broken down into the tokens “Straßen” (roads), “an Häusern” (near houses), “Stallungen” (stables) and “Düngerstätten” (manure heaps). This process created 475 habitat tokens. These tokens were then translated to English using native speakers of English, German, French and Dutch and for other languages a combination of Google Translate (translate.google.be) and the multilingual collaborative dictionary Wiktionary (wiktionary.org). To conduct the analysis it was necessary to reduce the number of habitat terms, which was done in two stages. The anglicized tokens were first simplified to closely related terms. Thus the terms “by foot of the city walls,” “along walls,” “under walls,” “mud walls,” “foot of walls,” “under walls,” “under a wall” and “foot of the church yard wall” were all replaced by “by walls.” This process reduced the number of habitat words to fifty. These fifty words were then arranged into logically related categories. Thus “by walls,” “by fences” and “by hedges” were grouped together under the term “boundaries.” This reduced the number of habitat categories to fifteen: animal waste, boundaries (including walls), coastal, disturbed and grazed land, dry & bare soil, habitation, hills, horticulture, industry, rail, roads, sand and rock, shipping, waste, wetland. A full list of the tokens contributing to each category is provided in Table S3. Throughout the process, the tokens were kept associated with the date: either the year the specimen was collected, observed or the year of publication. To analyse the use of habitat words in the collected corpus, the simplified habitat terms were pooled into 20 year periods from 1780 onwards. The proportional use of each habitat term was then calculated for each period. Statistical analysis was conducted in R (version 2.15.2) using generalized linear modelling with binomial errors, weighted with the number of tokens contributing to each pool. All models were checked for overdispersion using the ratio of the residual deviance and the degrees of freedom, but none were found to be overdispersed.

Analysis of distribution

Except for the rare occasions when coordinates where available with the specimen or observation, georeferencing was carried out manually according to best practise (Chapman & Wieczorek, 2006). Error radii for coordinates were not available for most records in databases, but they were estimated for the coordinates georeferenced in this study; however, they were not used to select data for the analysis. The average error radius was 11 km and the mode and median were both 10 km. C. vulvaria is a largely lowland species and errors in georeferencing of these magnitudes are insignificant for bioclimatic modelling at a global scale compared to the other inherent biases in these data.

Species distribution modelling was conducted using the BioVel Ecological niche modelling workflow and services (www.biovel.eu). The ecological niche modelling workflows were run on 6th Aug 2014. The workflow uses the Maxent method based upon Phillips, Dudík & Schapire (2004) and using the openModeller web service (de Souza Muñoz et al., 2011). Models were created using the default parameter and all 19 layers of the WorldClim global climate layers 10 arc min, version 1.4, release 3 (Hijmans et al., 2005).

Non-European observations used for modelling were only those locations where it was clear, either from the notes on the specimens or from Floras, that the species forms persistent population at these sites and was alien. If there was any doubt to the status, modern Floras were consulted to ascertain the persistence of the species in the area. Seventy observations were collected from locations outside Europe in southern Argentina, California, Chile, New Zealand, South Australia, South Africa, Tasmania, Tierra del Fuego (Argentina) and Victoria (Australia). The status of C. vulvaria in South Africa and New Zealand is not clear and therefore the data was not included. It is also believed to be native to Mongolia but only one observation was found. After selection, 42 observations from the naturalised range were used to model the range; however, weeding of duplicates during the workflow reduced the number to 32. The dates of these records were from 1863 to 2012, although 86% dated from 1950 onward. To model the realized range, all global observations were used, which resulted in 1894 observations after weeding of duplicates.

Results

Text analysis

Four habitat categories were notably more frequent than the others (Fig. 1). These categories are firstly waste, including rubbish piles, rubble, ruins and waste places of all kinds; secondly, boundaries, mainly at the base of walls; thirdly, roads and roadsides, including streets and farm tracks; and fourthly horticulture, such as gardens and other cultivated places. The habitat categories in Fig. 1 are not mutually exclusive, but often describe different aspects of the same habitat such as the proximity to landscape features, soil type, nutrient status and moisture.

Figure 1 The use frequency of words in the collected corpus of Chenopodium vulvaria habitat descriptions.

The frequency of each habitat category in the corpus of habitat descriptions from literature and specimens. The word and phrase tokens contributing to each category are presented in Table S3.

In summary, the habitat analysis underscores several aspects. C. vulvaria is strongly associated with humankind; natural habitats such as coastal habitat and wetlands are mentioned infrequently. C. vulvaria is intolerant of competition; none of its habitats are defined by established vegetation types, such as meadows, woodland or heaths. It is frequently associated with transport routes and it is usually associated with some form of soil disturbance.

When the use of these terms was compared over time, no significant change was found for the use of terms relating to animal waste, coastal, dry & bare soil, habitation, hills, horticulture, industry, rail, roads, shipping and waste. Figure 2 shows the changes of eight of these categories, including the only four where there were significant changes. The significant changes were increases in the proportion of the terms related to wetland (p < 0.01, DF = 11), sand and rock (p < 0.05, DF = 11) and disturbed and grazed land (p < 0.05, DF = 11), whereas there has been a significant decrease in the proportion of terms related to boundaries (p < 0.001, DF = 11). Of these significant changes only terms relating to boundaries were also highly frequent in the corpus (Fig. 1).

Figure 2 The change with time of habitat categories from the collected corpus of Chenopodiumvulvaria habitat descriptions.

A temporal analysis of the corpus of habitat descriptions from publications and specimens of Chenopodium vulvaria. The graphs show the proportion of token usage related to each habitat category for periods of 20 years. The words contributing to each habitat category are listed in Table S3. The best fit lines are from generalised linear models of the data weighted with the number of tokens contributing to each proportion. The categories wetland (P < 0.01, DF = 11), sand and rock (P < 0.05, DF = 11), and disturbed and grazed (P < 0.05, DF = 11) all significantly increased with time. Only the term boundaries decreased with time (P < 0.001, DF = 11). All other categories shown in Fig. 1 did not show significant variations with time.

Those 194 habitat tokens that also had a geolocation were plotted on a map to show the spread of habitat categories (Fig. S4). Habitat categories were widely distributed rather than being clustered in single counties or in areas of a common language. There is no indication of bias in the translation and the categorization of habitats. It also implies that the results of the habitat analysis are broadly applicable.

Introduction vectors, pathways and origins

Not surprisingly, clear expressions of the introduction vector were rare on specimens and in publications. Where introduction vectors were evident they are summarised in Table 1. Ballast soil at ports was the earliest vector mentioned in the corpus and it was also most frequently mentioned. However, it stops being mentioned in the early 20th century. Several specimens and observations implicate the transport of ore. C. vulvaria was reported on Chromite in Baltimore, USA between 1953 and 1958 (Reed, 1964), in Norway in 1954 (Uotila, 2001), on manganese ore in Norway between 1931 and 1935 and near an ore crushing plant in Kyrgyzstan in 1961 (Lazkov, Sennikov & Naumenko, 2014). Various agricultural products are mentioned as possible vectors of introduction, such as grain. However, there is no mention of it being introduced with other produce commonly imported from the Mediterranean such as herbal medicines or tobacco, even though C. vulvaria is frequently associated with waste.

Table 1 Introduction vectors gleaned from historical sources.

The vectors stated or implied from specimens and publications, including the range of dates that vectors were mentioned either on specimens or in publications.

Vector	Dates	Number	Example references and specimens	
Ballast	1870–1912	13	Publications: Burk, 1877; Mohr, 1901; Hjelt, 1906;
Specimens: BIRM 032912;
MANCH.94943.Kk803; S-H-2810	
Grain	1936–1964	3	Uotila, 2001; Unaccessioned specimens from
National Herbarium Nederland (L)	
Wool	1909	1	Observation by IM Hayward, Selkirkshire in database of the
Botanical Society of Britain and Ireland (2013)	
Ore	1931–1961	4	Uotila, 2001; Reed, 1964; Lazkov, Sennikov & Naumenko, 2014;
Specimen S-H-2141	
Cork	1956–1966	1	Uotila, 2001	
Horticultural imports	2008	1	Hoste et al., 2009	

Evidence for the pathways of introductions is scant, but shipping and railways are mentioned. Although roads are the most frequently mentioned transport system (Fig. 1), it is unclear if the presence of this species on roads relates to the introduction pathway or whether roads just provide suitable habitat.

Evidence for the origin of introductions is also slim, although where the origin is mentioned it is always from a country in the Mediterranean region (Uotila, 2001). There is no evidence of return introductions from naturalised populations outside Europe.

Comparing the fundamental and realized climatic niche

The observations of C. vulvaria within Europe are from an inseparable mixture of stable populations and casual occurrences. It is therefore impossible to validate a model for the fundamental climatic niche of C. vulvaria. For this reason I did not attempt to refine the output of the models by adjusting their default parameters or by eliminating climate layers. It is nevertheless informative to contrast models created from the known naturalised range outside Europe with the realized range within Europe (Figs. 3 and 4). The actual climatic niche, predicted from observations from the naturalised range outside Europe predicts the presence of C. vulvaria in southern and western Europe, North Africa and the Middle-east, notably, Spain, western France and Turkey (Fig. 3). The actual observations and the climate niche model created from them show a much wider distribution, which extends much further north and eastward than the niche model created from the naturalised range (Figs. 3 and 4). Figure 5 shows the locations of actual observations and the dates they were made. It demonstrates that there has been a general decline in the number of observations from northern Europe, but it also suggests unevenness in surveying effort between different countries and different time periods.

Figure 3 A distribution model created from the naturalised range of Chenopodium vulvaria outside Europe and extrapolated back to Europe.

A distribution model of Chenopodium vulvaria in Europe, North Africa and the Middle-east projected from its naturalised range in California, South America and Australia. This model aims to predict where, according to the naturalised range, the climate is suitable for persistent populations in Europe as opposed to casual occurrences. The map uses a Mollweide equal area projection.

Figure 4 A distribution model of Chenopodium vulvaria created from all known locations.

A distribution model of Chenopodium vulvaria in Europe, North Africa and the Middle-east created from all observations globally. This model aims to delimit the area where the climate is suitable for both stable populations and casual occurrences to occur. The map uses a Mollweide equal area projection.

Figure 5 A dated distribution map of Chenopodium vulvaria observations from Europe, North Africa and the Middle-east.

A distribution map of Chenopodium vulvaria in Europe, North Africa and the Middle-east. Circles represent georeferenced observations either from specimens or from the literature. The colour of the points denotes the date of observation, yet to emphasise the scarcer old records the date ranges are not equal, but the data are divided into equal-sized subsets. The map uses a Mollweide equal area projection.

Discussion

This study tracks the distribution and habitat changes of C. vulvaria over more than 200 years. Over this period botanical literature becomes more common and sufficiently abundant for analysis. Simultaneously, botanical specimens became more frequently collected and better documented, further adding to the analysable corpus of historical documents.

Over the past two centuries many social, economic and technological changes have occurred that may have influenced the abundance and distribution of C. vulvaria. Some key events in this period are: the expansion of the railway network in the 19th century; the adoption of motorised road transport in the early 20th century; the decline in the uses of horses for transport and agriculture in the 20th century; the transition from sail to steam powered ships at the turn of the 20th century; the discovery of herbicides in the mid-20th century; and the Green Revolution in the latter half of the 20th century. C. vulvaria to some extent benefits from its association with humankind; however, for the same reason it will be more acutely affected by cultural and technological changes than many other species.

Key habitat features of C. vulvaria were identified though text analysis. This species has been, and still is, strongly associated with humankind, both as a weed of cultivation and as a ruderal plant. The analysis identifies habitat traits such as its avoidance of competition and the association with waste. The genus Chenopodium is considered to be nitrophilous (Ellenberg et al., 1991); indeed C. vulvaria is sometimes associated with habitats linked to animal dung, however, it is much more commonly associated with other types of waste or cultivated place (Fig. 1).

The temporal analysis of habitat change indicates that C. vulvaria is still associated with many of the same habitats as it was in the past, such as agriculture, transport and waste (Fig. 2). However, in the 20th century habitat descriptions have included proportionally more words related to natural or semi-natural habitats, such as grazing, sand and wetland.

The reference to wetland amongst the habitats needs further explanation, because C. vulvaria is not a typical wetland plant. It does not grow in water, but colonises bare soil exposed in the summer at the margins of rivers, ditches and lakes. Thus, its association with wetland is of an opportunistic colonizer of habitats free from competition, rather than a true wetland plant.

The habitat where C. vulvaria has declined is along boundaries, particularly along walls, which contributed 80% of the boundary terms. C. vulvaria does not grow on walls but beside them, which appears at first sight to be a rather non-specific habitat description. However, the margins of walls have changed considerably in the past 200 years. Walls were once built using lime mortar, rather than cement, and were frequently painted with whitewash, a mixture of calcium hydroxide and chalk. Whitewash gave the traditional white or pink colour to houses throughout Europe. Consequently, the soil in the immediate vicinity of walls would have been alkaline. C. vulvaria is not known as an alkaliphile, however it is clearly tolerant of high pH as it has been found on the ultrabasic rock chromite (Reed, 1964). Furthermore, because horses were used for transport and farm animals were driven along roads, the base of walls would have been strewn with animal waste. Such fertile alkaline habits do not occur by walls in modern towns and it is speculated that the technical changes in building practises and changes to transportation have contributed to the decline of C. vulvaria.

Text analysis is clearly a useful tool for environmental historians; nevertheless, it is susceptible to the fallibility and biases of authors, who may uncritically follow their forbears or write from hearsay rather than experience. Also, botanical activity is spatially and temporarily biased (Delisle et al., 2003; Schmidt-Lebuhn, Knerr & Kessler, 2013). For example, British and German botanical literature has, and continues to be, more abundant than for other countries in Europe.

Compared to the analysis of habitat, evidence for introduction vectors, pathways and origins was limited. The results, however, suggest that there were multiple vectors introducing C. vulvaria to northern Europe, but particularly as a grain contaminant and in ship’s ballast. The frequent occurrence of C. vulvaria in waste perhaps indicates that its seeds were contaminants of many crops. Indeed, different descriptions accompanying specimens mentioned C. vulvaria in crops of lentils and potatoes. Unfortunately, the source of a casual introduction is rarely obvious by the time the plant is mature. Weed species that are dispersed as seed contaminants have declined throughout Europe in the 20th century; this is, in part, a consequence of improved seed cleaning methods (Hilbig, 1987; Sutcliffe & Kay, 2000; Lososová, 2003). Most of these weeds are considered archaeophytes to northern Europe; typical examples are Agrostemma githago L. Glebionis segetum (L.) Fourr. and Lithospermum arvense L.

Soil was used as ballast on sailing ships during the 18th and 19th centuries to provide stability to cargo ships when not carrying heavy loads. In ports, where heavy materials were loaded, ballast was removed and replaced by cargo. Large hills of ballast soil were a common feature of busy ports, particularly in areas of mining and heavy industry, such as in northern Europe. These ballast hills were a large reservoir of propagules for many species (Carlton, 2011). The large number of specimens and observations reflects the importance of this invasion pathway, but might be somewhat over-represented because botanists were attracted to ballast heaps as a source of novel species and because the vector of the propagules is clear in this case.

Ore is also mentioned as an introduction vector to the USA and Norway. Chromium processing began in Baltimore, USA in 1822, at which time only local chromium ore deposits were processed (Newcomb, 1994). However, by the end of the 19th century local chromium deposits were exhausted and processing continued with imported ore until the end of the 20th century. Similarly, Norway is also a large processor of imported chromium ore; for example, in 1992 the country imported 187,965 tonnes of chromite ore from Turkey (Plachy, 1992). Indeed, it is likely that some of the chromite imported into Baltimore was also from Turkey where chromite was first mined in the 19th century (Zengin, 1957). Therefore, it seems that exports of chromite from Turkey could have been a pathway for dispersal of C. vulvaria during the 20th century.

Animal dung is often mentioned as a growing medium for C. vulvaria, which is indirect evidence for endozoochory. Certainly, other Chenopodium species are dispersed in this manner and C. vulvaria is eaten by ruminants despite its smell (Withering, 1776; Haarmeyer et al., 2010). In the 21st century yet another vector of C. vulvaria introduction has been created, that of imported olive trees (Hoste et al., 2009). These mature trees are extracted from olive groves with a large amount of soil and are sold in northern Europe as horticultural novelties.

Though dispersal vectors are rarely mentioned in the corpus, it is clear that C. vulvaria has been dispersed by a wide variety of vectors and through a number of pathways (Table 1). There are historic periods associated with each vector and if this analysis was extended to more species, one would be able to further refine the time frames during which these pathways were operating. From the diversity of distribution vectors it is clear that C. vulvaria has been widely introduced outside its natural climatic range and it often grows temporarily. However, with the exception of horticultural imports, introduction pathways of C. vulvaria declined midway through the 20th century.

The sporadic occurrence of C. vulvaria presents a problem for the selection of occurrences for distribution modelling. Unless all casual occurrences are eliminated from the data before fitting, the model would indicate a much broader climatic range. Separating permanent populations from casual occurrences is impossible for Europe, where anthropogenic disturbance and trade have confused the quasi-natural distribution. However, in the naturalised range the situation is much clearer. Most, if not all, modern observations of C. vulvaria in California, Australia and South America appear to be from naturalised populations, that is to say, the associated meta-data indicates the presence of a population, and there is no indication of a recent introduction. Therefore, the naturalised distribution outside Europe should reflect the fundamental climatic niche of the species, as long as the distribution is at equilibrium. This assumption seems reasonable since old casual records of C. vulvaria occur throughout the world, but naturalised populations persist in only a few of those places. Clearly, introduction events were occurring all over the world for several hundred years of international trade, but C. vulvaria only naturalised in a few of those places where habitat and climate were suitable.

Projecting the bioclimatic range in Europe from naturalised alien populations elsewhere predict a more south-western distribution of C. vulvaria than the modelling using all occurrences (Figs. 3 and 4). However, these rather crude models indicate that the naturalised distribution of C. vulvaria has a climate much closer to that of southern Europe and North Africa than to northern and central Europe. The distribution models are consistent with my hypothesis that historically C. vulvaria was only present in parts of northern Europe because of repeated introductions, and that, in these places, the climate is unsuitable for lasting populations to exist.

This conclusion assumes niche conservatism between the native and introduced populations. It further assumes that the whole of the fundamental niche climate space is available to C. vulvaria in its introduced range (Soberón & Peterson, 2011). Therefore, an alternative hypothesis is that C. vulvaria has undergone a climatic niche shift in its introduced range such as suggested by Gallagher et al. (2010). However, given that this would have had to occur in parallel on three continents, this seems less plausible.

Discrepancies between the projected model and the realized distribution could be the result of several factors: an incorrect model; lack of suitable habitat; spatial variation in surveying efforts; or plants growing outside their actual climatic niche due to local factors. The model projecting distribution from the naturalised range is based on relatively few observations and would be improved by more data (Feeley & Silman, 2011). Nevertheless, any distribution model of this species has to address the problem of casual occurrences. The shortage of observations from countries such as Turkey and Morocco, in apparent contrast to the models, is at least in part due to lack of collecting in these regions, but also due to the inaccessibility of the data from these countries. These results are a good reminder to those who would extrapolate native ranges onto potentially invasive ranges that it is not always possible to predict the naturalised distribution of an invasive species based upon fundamental climatic niche both because of the lack of data and indistinct native range boundaries, but also because climate is not the only determinate of distribution.

Conclusions

Text analysis is a useful technique to study recent ecological and distributional change. Despite its limitations, it provided information, which would be difficult if not impossible to obtain from other sources. As a larger volume of semantically enhanced biodiversity literature becomes available it will allow much more sophisticated habitat analysis covering many more species. The ability to contrast data from different species will strengthen results and allow correction for some of the biases. Furthermore, the development of environmental ontologies and thesauri will simplify the method and improve repeatability (Buttigieg et al., 2013). This will allow over-representation analysis of ontological terms associated with one species compared to the incidence of these terms in the whole corpus.

Analysis of environmental descriptions indicates that the habitat of C. vulvaria has changed over the past two centuries, particularly next to walls. Multiple vectors and pathways have been involved in the human mediated dispersal of C. vulvaria, but different vectors and pathways were active in different periods. In the past C. vulvaria would have been dispersed to many places outside of its climatic niche. It is reasonable to believe that many of the observations of C. vulvaria in northern Europe were the result of introductions and that a reduction in the propagule pressure in recent years has consequently lead to a decline in observations of this species. It is concluded that humankind spread C. vulvaria to northern Europe and created habitat for it to grow and then inadvertently removed the habitat and the introduction pathways causing a decline.

Supplemental Information

File S1 A reference list to the sources of data

Citations of the sources of locations and habitat descriptions of Chenopodium vulvaria.

Click here for additional data file.

File S2 Digital data sources

Citations of data providers from the Global Biodiversity Information Facility and the Atlas of Living Australia.

Click here for additional data file.

Table S3 A summary of the habitat description simplification terms

The words and phrases from specimens and Floras contributing to each broad habitat category.

Click here for additional data file.

Figure S4 A map of the four most common habitat categories

For the four most common habitat categories where specimens or observations had both a habitat category and a geolocation they have been mapped to show the distribution of habitat categories within Europe. Points have been randomly jittered to ensure overlapping points are visible. The map uses a Mollweide equal area projection.

Click here for additional data file.

I would like to thank the following people for their help accessing collections and transcribing specimen details: Ana Isabel D. Correia of the Faculdade de Ciências da Universidade de Lisboa; Prof. Dr. Roberto Rodríguez Ríos from the Universidad de Concepción, Chile; Noemí Montes and Neus Ibáñez Cortina of the Institut Botànic de Barcelona; Sabrina Eckert, Anton Güntsch, Patricia Kelbert, Andreas Müller and Robert Vogt of the Botanischer Garten und Botanisches Museum Berlin-Dahlem; Vladimir Vladimirov from the Bulgarian Academy of Sciences; Michal Štefánek of Charles University in Prague; Alan Paton from the Royal Botanic Gardens, Kew; Leni Duistermaat & Dr. Hubert Turner of Naturalis Biodiversity Center and the specimen digitization team at the Botanic Garden Meise and Sabine Metzger.

Additional Information and Declarations

Competing Interests

Author Contributions

Data Deposition

Quentin J. Groom is an employee of Botanic Garden Meise and a volunteer with the Botanical Society of Britain and Ireland. He is on the Editorial Board of the New Journal of Botany and a Subject Editor for the Biodiversity Data Journal.

Quentin J. Groom conceived and designed the experiments, performed the experiments, analyzed the data, contributed reagents/materials/analysis tools, wrote the paper, prepared figures and/or tables, reviewed drafts of the paper.

The following information was supplied regarding the deposition of related data:

http://www.gbif.org/dataset/28ac18b4-8c7e-4873-89af-86fcf35b118b.

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
