# Peer review of "Piecing together the biogeographic history of Chenopodium vulvaria L. using botanical literature and collections"

_PeerJ, doi:10.7717/peerj.723_

## Round 0.1 · original submission · Major Revisions

· Academic Editor

Major Revisions

This paper uses an interesting approach to reconstruct historical distribution and habitat. It is the approach that is the selling point for this paper, and for that reason I have indicated major revision, as the issues raised by reviewers 2 & 3, linked to your "reversed" distribution modelling approach, require strong substantiation; if your approach is novel, then more should be made of it in the paper. Reviewer 3 suggests a look at invasion-related literature, where similar approaches, comparing native and invaded species ranges, have been used; I would like to see some engagement with this literature in the paper. Please also consider, and respond to, Reviewer 2's suggestions to include a latitudinal aspect in your GLM, and to justify sample size. Reviewers have highlighted some more minor editorial suggestions in annotated uploads of your paper; please also search for the word "data" and ensure you reflect the plural nature of the word.

·

Basic reporting

Well written. No comments except that in manuscript acknowledgements follows References whereas in PeerJ it should be the reverse.

Experimental design

No comments

Validity of the findings

No comments

Reviewer 2 ·

Basic reporting

Some very minor grammar/spelling errors and recommendations for figures - please see attached manuscript.

Experimental design

I understand that the study design is limited by available material, but 32 observations seems very low to confidently base distribution modelling on. I believe the equilibrium assumption of the modelling is adequately met by controlling for well-established and long-term extra-European populations, as the author does, but by how much does this decrease the extra-European sample size?

It would be interesting to see whether the observed trends in habitat description differ between the north and south of Europe, and would be directly relevant to the study. Even if this is done on a very rough scale (northernmost 50% of data points vs southernmost 50% as a categorical variable), this might be able to tease apart differences in habitat preference between putative natural and introduced distributions and would be easy to include as a second factor in the glm.

Validity of the findings

I think it would add significantly to the manuscript if the statements in the introduction on the declining nature of C. vulvaria in the northern parts of its range were to be substantiated with the impressive data set that the author has gathered. The entire argument of the paper is based on this premise, and should be very easy to test, say as a declining trend in a regression of latitude over time, controlling for temporal collecting effort (as the author has already partially done in figure 5) and possibly spatial collecting effort. The reason for this is that it is not at all evident to me in figure 5 that C. vulvaria has declined in the north of it's range - the only visual trend possibly evident from figure 5 is increased clustering of collecting localities over time (compare the fairly evenly distributed earliest collecting localities with the most recent, which are overwhelmingly distributed close to Paris, Athens and London). Not to say that there is no decline, only that it is of obvious relevance to the paper that the data are consistent with one of its major premises.

Comments for the author

This paper is significant for two reasons, namely the laudable argument for the immense potential value of digitised legacy literature and the novel (to me, at least) twist on species distribution modelling, using naturalised populations in introduced regions to map potential original distributions instead of the reverse. It is in general well-written and clearly argued, and the care and effort with which the author has assembled this substantial database is impressive.

I have two issues that I feel need to be addressed before acceptance for publication, namely 1) the sample size for the naturalised range and 2) confirmation that the claim for a declining trend in northern Europe is real.

1) I have very limited experience with species distribution modelling, but the sample size for introduced range seems excessively small. I think the issue here is a Catch-22 between meeting the model assumptions (i.e equilibrium) and adequate sample size to make any meaningful deduction. I don't know how many total data points were present in the introduced range before excluding non-naturalised population accessions, but if this is a substantial number it might be useful to include them and see whether the inferred original range still excludes northern Europe.

2) this speaks to the heart of the paper, and would be very nice to have confirmed by the present data set. If confirmed, then great, but if not, then some explanation is necessary for the discrepancy. This can also be further confirmed by including some latitudinal aspect in the glm (for Europe, specifically) - it would be extremely interesting and obviously relevant to link a potential general decline to a decline in a particular habitat type/introduction pathway in the north vs the south.

I am recommending major revision for this manuscript because of the potential impact both these issues can have on the paper.

·

Basic reporting

This is a well designed and well written paper on a rather unusual way of piecing together historical and current species distribution data. I have very few comments to make on this paper as it scientifically sound and very well written.
I only battled a bit with what I assume is the abstract. On first read I found the abstract confusing and only after reading the paper did it make proper sense to me. In the abstract the author first states that the native range of Chenopdium vulvaria is southern Europe and later on states that it may also have been native to northen Europe. From the paper it becomes clear that the historical native range of this species is not exactly known. From this analysis it can be inferred that the historical range was probably southern Europe although northern Europe cannot be convincingly excluded as part of the native range. In the abstract this logic is not fully represented.
Since the abstract is the first part and perhaps the most important section a potential reader would look at, the author should try to avoid any ambuigity in the text. Some rephrasing and clarification of the abstract is recommended.

Experimental design

Experimental design and analysis is sound and I have no further comments on this.

Validity of the findings

Because of the nature of the data, the findings will remain speculative to a certain extent. Nevertheless, the analysis and the findings are valid and exciting. In my opnion, the strongest point of the this paper, although it is not highlighted as such, is what I would call the reversed species distribution modelling approach. On the basis of the current naturalised distribution of this invasive species, the author makes inferences about the historical native distribution of the species, which is not exactly known. I think this is quite an interesting approach. However, as I am not familar with the literature on this particular topic, I would like to know if anyone else has used this approach and if this approach has been validated for instance for invasive species for which the historical distibution is well established. It would be good to have some reflection on this in the discussion. If this technique has not been used by others, then I would consider the approach of the authors quite novel, and I would encourage him to flesh this out a bit more in depth.

Comments for the author

A fine paper.

·

Basic reporting

yes, meets the basic reporting requirements

Experimental design

-Author clearly lays out research question
to prove a hypothesis "C. vulvaria was formally more abundant in northern Europe and its current decline is the result of changes in the introduction pathways and loss of habitat. "
-Author clearly explains their methodology for which data sources they searched and the tools they used for analyzing the data in systematic way.
-Approach was novel in dealing with sources across numerous languages by breaking terms down into habitat tokens and focusing on the translation of just those words

Validity of the findings

-Data is statistically sound and appropriately controlled.
- Data sources are clearly identified
-Author clearly identifies points of weakness in research where data sources were non- existent for certain geographic regions or biased
-Author ties distribution models nicely back to the original hypothesis in the conclusion!

Comments for the author

Wonderful paper- enjoyed reading it very much! Just a minor correction on page 4 " Observations where gathered from the biodiversity literature by reading the BHL corpus systematically after searching for Chenopodium vulvaria L. and its synonym Chenopodium olidum Curt. " change "where" to "were"

---

## Round 0.2 · Minor Revisions

· Academic Editor

Minor Revisions

Thank you for your detailed response. I have added a few comments to your document in tracked changes; this document will be mailed to you.

---

## Round 0.3 · accepted · Accept

· Academic Editor

Accept

Thank you for the revised version of your paper which is now suitable for publication in PeerJ.